# Mindfulness, Subjective Cognitive Functioning, Sleep Timing and Time Expansion during COVID-19 Lockdown: A Longitudinal Study in Italy

Marco Fabbri

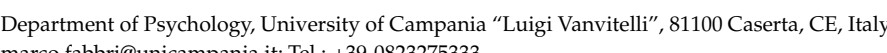

Department of Psychology, University of Campania "Luigi Vanvitelli", 81100 Caserta, CE, Italy; marco.fabbri@unicampania.it; Tel.: +39-0823275333

**Abstract:** During the COVID-19 lockdown, a distortion of time passage has been widely reported in association with a change in daily rhythm. However, several variables related to these changes have not been considered. The purpose of the present study was to assess the changes in dispositional mindfulness, time experience, sleep timing and subjective memory functioning. A longitudinal study was conducted on 39 Italian adults (53.85% males; 35.03 ± 14.02 years) assessing mindfulness, ad hoc questions of sleep habits during workdays and free days, chronotypes, subjective time experience, and memory functioning before (December 2019–March 2020) and during (April 2020–May 2020) the first Italian COVID-19 lockdown. Participants reported delayed sleep timing, a slowdown in the perception of the present time, a decrease of time pressure, and an increase in the feeling of time expansion/boredom. In addition to correlations between mindfulness, memory functioning, and subjective sleep duration during workdays, a mediation model showed that changes in the dispositional mindfulness determined a delay of bedtime during workdays through the mediation effect of increased feeling of time expansion/boredom. This finding highlighted the role of mindfulness in reducing the feeling of time expansion/boredom for regulating the sleep timing. The theoretical and practical implications of the findings are discussed.

**Keywords:** dispositional mindfulness; sleep timing; sleep continuity; retrospective memory; prospective memory; subjective time awareness; COVID-19 lockdown; longitudinal study; circadian typology; boredom

## 1. Introduction

In December 2019, from the Wuhan province of China, the coronavirus disease 2019 (COVID-19) spread rapidly worldwide and was caused by severe acute respiratory syndrome—or coronavirus2 (SARS-CoV2) [1]. Thus, the World Health Organization (WHO) declared COVID-19 as a Public Health Emergency of International Concern (PHEIC) and the pandemic state [2,3].

Several national governments in almost all countries around the world imposed a strict social distancing measure, with the closure of most business and recreation activities (i.e., non-essential commercial activities), as well as the shutdown of school and group meetings [4]. This lockdown impacted worldwide on individual wellbeing and societies [5]. For example, it has been widely reported that the social isolation, due to lockdown, produced negative psychological effects on the general population in different countries, such as China, Spain, Italy, and the United States [6–9]. Specifically, people experienced anxiety, depression, stress, poor sleep–wake quality, and cognitive dysfunction [10–17].

In addition, a change in the experience with time has been reported during the home confinement. Generally, the studies have reported a slowing down of time during the home confinement [18–30]. In other words, people experienced a slowdown in their passage of time, and this time distortion seemed to be related to social isolation, negative emotions, boredom, and distress. Specifically, several studies indicated that the boredom experienced

during home confinement was strictly associated with the distortion (i.e., slowdown) in the passage of time [18–30]. Related to boredom, in a recent study with Chinese college students, the authors found that, during the COVID-19 outbreak, the boredom proneness was negatively associated with self-control and positively associated with bedtime procrastination [31]. At the same time, it has been found that bedtime procrastination was a predictive factor for sleep deficiency during the COVID-19 outbreak [32]. Thus, it is possible to advance the idea that a distortion in the passage of time could be related to sleep timing and regularity during the pandemic social isolation.

The proposed idea could be grounded on the evidence, during the COVID-19 lockdown, of a change in life rhythm [4,19,21], which was associated with a change in the feeling of the passage of time [18,21]. Considering that sleep timing and regularity were affected by the lockdown [31,32], it is possible to posit that people who are bored (i.e., people who experience a negative mood associated with a lack of a satisfying activity) have difficulty to follow a specific daily schedule, due to a slowdown in the passage of time and a lack of a life rhythm, should impact on sleep quality. At the same time, people with a poor night's sleep report less energy to do something in the following day and are more bored, experiencing a slowing down of time passage [18,19,23,25–29]. In line with these predictions, Cellini et al. [18] reported an association between an increase in the experience of time dilatation and worse sleep quality, as well as Martinelli et al. [26] showed that sleep difficulties were one of the main predictors of the feeling of a slowing down of time. However, Droit-Volet et al. [19,25] did not find a significant predictive effect of sleep quality on the change in the subjective experience of the passage of time, even when people returned to a more regular life rhythm with a better quality of sleep, but a persistent feeling of boredom remained. These mixed results could be related to the fact that these studies did not directly investigate other sleep characteristics such as sleep timing (e.g., bedtime and wake-up time, midpoint of sleep [33]) and sleep regularity (e.g., social jetlag [34]), which impact important characteristics of sleep health [35]. Indeed, it could be advanced the idea that bored people seek activities that stimulate them and are distracted from sleep in an attempt to find something to do [31,32]. Thus, boredom could predict bedtime procrastination because bored people give in to temptation, delay sleep timing, and have difficulty in maintaining a regular sleep–wake pattern. The present study aimed to investigate whether the boredom and a dilatation of the passage of time could alter sleep timing and regularity during the COVID-19 outbreak.

However, to better address the aim of the present study, the circadian typology (i.e., main individual differences in chronopsychology [36]) should be considered. Circadian typology can be seen as a continuum of different chronotypes, from morning-types to evening-types, passing through intermediate-types [36]. These chronotypes differ for their ability to estimate time, given that it has been reported a general underestimation for morning-types and a general overestimation for evening-types [37,38]. When time perspective (i.e., how individuals perceive and relate to the past, present, and future) is considered, it has usually reported that evening-types are more oriented towards a present-hedonism, while morning-types are more oriented towards a positive view of the future [39–44]. When the subjective feeling of the passage of time has been addressed, Beracci et al. [44] found that evening-types attributed higher scores to future temporal expressions (i.e., a sort of index of procrastination) and overestimated the passage of time. Thus, evening-types seem to report a general slower sense of the passage of time. In addition to this difference, it has been repeatedly reported that morning-types usually go to bed and wake up early whereas evening-types usually go to bed and wake up late [36]. This different sleep–wake pattern suggests that morning-types are more adapted with social routines that are morning-oriented [34], whereas evening-types are more likely to experience a mismatch between their internal biological clock (i.e., their spontaneous circadian preference) and external social schedules (i.e., work or school timing). Thus, evening-types report a greater social jetlag and sleep restriction (i.e., the shift in sleep duration between workdays and free days [45]) because their sleep episode is shortened due to a delay in the sleep

onset and an early awakening induced to comply with the social times (e.g., school/work start time). Thus, evening-types tend to experience a shorter sleep duration compared to morning-types, especially during working days [36]. During the COVID-19 pandemic, several studies showed that evening-types reported the most prevalent delay of the sleep phase [46–48] and lower sleep quality, highest level of insomnia, depression, perceived stress, and anxiety, than those reported by morning-types [47]. Thus, it is possible to expect that evening-types should report a dilatation of time, high boredom, a delay of their bedtime, and less sleep regularity.

In the present study, the dispositional mindfulness (or sometimes called trait mindfulness) was considered as an additional individual difference. Trait or dispositional mindfulness could be defined as a personality trait and refers to the innate capacity of paying and maintaining attention to the present moment with a nonjudgmental attitude [49,50]. The dispositional or trait mindfulness could be measured with subjective self-report questionnaire, and the Mindful Attention Awareness Scale (MAAS) [51] is a promising (and widely used) self-reporting measure [52], targeting the attention to and awareness of the present moment. A possible reason why it is useful to consider the dispositional mindfulness could be found in the assumption that bedtime procrastination could be related to ignoring the present moment (a component of mindfulness) in the attempt to seek to do something of interesting [53]. This assumption could be based on the evidence that mindfulness practice induced an improvement of individual attention-controlling abilities [54–61], with better accuracy in discriminating different time scales (e.g., seconds or minutes). Moreover, it has been reported that the practice of mindfulness meditation induced a faster judgement of the passage of time than other control exercises or no meditators do [62–64]. Thus, it is possible to suggest that people with higher dispositional mindfulness should be more focused on the present moment and less on the past and the future, in line with mindfulness practitioners [62–64]. Furthermore, it is possible to posit that high dispositional mindfulness should induce a faster time experience (or less boredom), and, consequently, an earlier sleep timing and regularity in sleep patterns with consistent bedtimes and wake-up times, even during the social isolation. From one hand, several studies have reported an association between mindfulness and sleep quality/health, even during the COVID-19 lockdown [65–67], and on the other hand, Teoh and Wong [68] have recently found, in a cross-sectional study in young adults, that mindfulness predicted a lower level of boredom, which, in turn, was associated with a lower level of bedtime procrastination, and, consequently, better sleep quality. However, previous studies did not assess the passage of time and sleep–wake timing and/or regularity, and the present study tried to fill this limit.

Finally, in the present study, the memory functioning was considered because memory involves a retrospective component (to remember past events, e.g., "what I have done the past weekend") and a prospective component (to remember to perform an action in the future, e.g., "I have to remember to bring the book to my classmate when I will see him") [69,70], and the optimal functioning of both components is crucial in everyday life to successfully complete a wide range of daily activities. Furthermore, in prospective memory, there is the distinction between event-based (i.e., to perform an action in response to a specific external cue) and time-based prospective (i.e., to perform an action either at a specific future time or after a certain amount of time has passed) tasks [71], suggesting a further link with time perception. The temporal components of the memory could be related to retrospective timing (i.e., participants are not informed in advance that they will make a time-related judgment) and prospective timing (i.e., participants are informed in advance that they will make a time related judgment) reported in the temporal domain [72,73]. The retrospective timing is influenced by the number of memories retrieved, whereas the prospective timing is influenced by amount of attention requested by the task. It is worth noting that the attentional–gate model proposed by Block and Zakay [74] could explain the performance in temporal tasks as well as in memory tasks. Indeed, this model posits a pacemaker mechanism, which produces temporal pulses passing through an attentional–gate and into an accumulator. Thus, prospective (time) estimations arise from a cognitive

comparison between the number of pulses accumulated and pulse count information stored in a long-term reference memory. In a similar way, during a time-based prospective task, temporal pulses are collected into the accumulator, given that the participant should attend to the passage of time to perform the prospective task accurately. When the number of pulses collected matches that of the time-based target response time, the internal clock signals that it has reached the time to perform the prospective response [74]. In addition to the relationship between memory and time perception, in recent years, it has been widely shown as a sleep effect on both types of memory, although the relationship between sleep and prospective memory is still under debate. Interestingly, a recent study [75] speculated that the lower accuracy in executing prospective memory at bedtime could be related to the higher pressure of the homeostatic sleep regulation process [76] and decreased alertness levels. Thus, it is possible to expect that people with altered sleep timing and regulation, as well as lower dispositional mindfulness, should be impaired in memory functioning during the home confinement. Indeed, it has been reported that home confinement impacted on subjective cognitive complaints, such as executive function and memory [12,13,16,17,77]. Using the Prospective and Retrospective Memory Questionnaire (PRMQ) [78], which is usually used to assess memory slips in daily life for both retrospective and prospective components, Fiorenzato et al. [12], however, found a paradoxical effect for memory, with a general improvement of memory abilities during the first COVID-19 confinement. Although the authors reported an increase of the sleep problems and complaints in attention, temporal orientation, and executive functions, they did not explore the associations between the PRMQ score and these variables, and the present study tried to tap this limit.

In the present research, the general aim was to assess the relationships between the subjective passage of time, sleep timing and regulation, chronotype, dispositional mindfulness and self-reported memories slips in a longitudinal study (from December 2019 to May 2020) in Italy. Contrarily to the majority of the studies, which required participants to compare, for example, their sleep quality during the COVID-19 lockdown with that usually exhibited in the pre-lockdown months, the present study could directly assess the changes experienced from pre-lockdown to first Italian lockdown in the same individuals for all variables considered. Thus, first of all, it was expected in all participants a slowdown of the subjective passage of time, longer sleep duration (i.e., reduced social sleep restriction), delayed bedtimes, reduced social jetlag, lower dispositional mindfulness and a reduction of self-reported memory slips. In addition, a slower passage of time, more irregularity in the sleep–wake habits and lower dispositional mindfulness should be expected in the evening-types, with respect to the other two chronotypes, while no chronotype differences should be expected for subjective memory functioning. Second, negative associations between reduction of the dispositional mindfulness and increase of dilatation in the passage of time, later bedtime, increase of sleep irregularity, and better memory functioning were expected. At the same time, a slowdown in the passage of time should be associated with a later bedtime, reported sleep irregularity and better memory functioning. Finally, the following mediation model was expected: the changes in the dispositional mindfulness from pre-lockdown to during COVID-19 pandemic outbreak should predict the changes in the feeling of time expansion/boredom, which, in turn, should predict the bedtimes/social jetlag of participants. Specifically, it was expected that people with a reduction of their dispositional mindfulness during home confinement should report a greater slowdown of their passage of time, and then a delayed sleep timing and less regularity.

## 2. Results

The comparison between the mean rMEQ score at pre-lockdown (M = 14.33; SD = 4.02) and that at COVID-19 lockdown period (M = 14.33; SD = 4.13) was not significant ($t(38) = 0.0001$, $p = 1.00$). In the sample, there were 10 evening-types, 21 intermediate-types and 8 morning-types. These chronotypes did not differ for gender ($\chi^2(2) = 1.10$, $p = 0.58$), and the rMEQ score did not correlate with age ($r = +0.24$, $p = 0.15$) or education level ($rho = +0.16$, $p = 32$).

During the COVID-19 lockdown, participants shifted in advance their bedtimes and wake-up time during workdays, and their MPoS was significantly advanced (Table 1). Additionally, participants reported a reduction of their feeling of time pressure and an increased feeling of time expansion/boredom (Table 1). In regard to the circadian typology effect, there was a slight trend towards the significance for sleep time during free days, reflecting their different biological rhythms. Evening-types tended to report a preference to go to sleep later with respect to that reported by the morning-types, and with the intermediate-types in the middle. No significant interactions were found for any variables ($Fs < 2.88$, $ps > 0.07$, and $\eta^2{}_p < 0.14$).

**Table 1.** For all samples, the means (and their relative SDs) of each variable for evening-, intermediate-, and morning-types for both pre-lockdown and during lockdown period, are reported. The $F$, $p$, and partial eta-squared ($\eta^2{}_p$) for circadian typology and time factors are also shown. The significant results and the tendency towards significance are shown in bold and in italics, respectively.

| Variable | Chronotypes | Pre-Lockdown | | During Lockdown | | $F$ | $p =$ | $\eta^2{}_p$ |
|---|---|---|---|---|---|---|---|---|
| | | **M** | **SD** | **M** | **SD** | | | |
| MAAS score | Evening-types | 62.00 | 14.67 | 60.40 | 16.28 | | | |
| | Intermediate-types | 59.38 | 9.11 | 56.71 | 10.47 | 0.26 | 0.77 | 0.01 |
| | Morning-types | 63.63 | 13.58 | 56.38 | 19.91 | | | |
| | Mean Survey Time | 61.67 | 12.45 | 58.83 | 15.53 | 4.11 | 0.05 | 0.10 |
| Retrospective score | Evening-types | 30.60 | 8.68 | 30.80 | 10.60 | | | |
| | Intermediate-types | 31.19 | 9.49 | 31.19 | 8.36 | 0.06 | 0.94 | 0.003 |
| | Morning-types | 27.75 | 13.60 | 32.00 | 13.05 | | | |
| | Mean Survey Time | 29.89 | 10.59 | 31.33 | 10.67 | 1.25 | 0.27 | 0.03 |
| Prospective score | Evening-types | 32.60 | 10.06 | 35.40 | 16.12 | | | |
| | Intermediate-types | 33.81 | 7.74 | 35.00 | 10.52 | 0.13 | 0.88 | 0.007 |
| | Morning-types | 35.00 | 10.74 | 37.38 | 14.07 | | | |
| | Mean Survey Time | 33.80 | 9.51 | 35.93 | 13.57 | 1.44 | 0.24 | 0.04 |
| Total PRMQ score | Evening-types | 30.80 | 9.40 | 32.50 | 13.93 | | | |
| | Intermediate-types | 31.71 | 8.55 | 32.52 | 10.00 | 0.01 | 0.99 | 0.001 |
| | Morning-types | 30.38 | 12.97 | 34.13 | 14.40 | | | |
| | Mean Survey Time | 30.96 | 10.31 | 33.05 | 12.78 | 1.99 | 0.17 | 0.05 |
| **BedTime Workdays (hh:mm)** | Evening-types | 01:32 | 01:13 | 03:24 | 02:51 | | | |
| | Intermediate-types | 24:16 | 01:16 | 01:43 | 02:33 | 2.45 | 0.10 | 0.12 |
| | Morning-types | 23:03 | 01:41 | 02:41 | 05:02 | | | |
| | **Mean Survey Time** | **24:17** | **01:21** | **02:36** | **03:29** | **16.32** | **0.0001** | **0.31** |
| **Wake-UP Time Workdays (hh:mm)** | Evening-types | 09:12 | 01:52 | 11:22 | 03:05 | | | |
| | Intermediate-types | 08:25 | 01:52 | 09:43 | 02:39 | 1.72 | 0.19 | 0.09 |
| | Morning-types | 06:30 | 01:04 | 10:57 | 04:41 | | | |
| | **Mean Survey Time** | **08:02** | **01:36** | **10:41** | **03:28** | **20.03** | **0.0001** | **0.36** |
| BedTime Free days (hh:mm) | Evening-types | 02:36 | 01:32 | 02:15 | 01:47 | | | |
| | Intermediate-types | 01:58 | 01:46 | 01:47 | 01:33 | *3.76* | *0.033* | *0.17* |
| | Morning-types | 24:53 | 02:11 | 24:08 | 01:33 | | | |
| | Mean Survey Time | 01:49 | 01:50 | 01:23 | 01:38 | 2.46 | 0.13 | 0.06 |

**Table 1.** *Cont.*

| Variable | Chronotypes | Pre-Lockdown | | During Lockdown | | $F$ | $p =$ | $\eta^2 p$ |
|---|---|---|---|---|---|---|---|---|
| | | M | SD | M | SD | | | |
| Wake-UP Time Free days (hh:mm) | Evening-types | 10:54 | 02:01 | 10:30 | 03:25 | | | |
| | Intermediate-types | 10:25 | 02:05 | 09:34 | 02:32 | 1.07 | 0.36 | 0.06 |
| | Morning-types | 09:21 | 02:14 | 09:04 | 01:43 | | | |
| | Mean Survey Time | 10:13 | 02:07 | 09:43 | 02:34 | 1.96 | 0.17 | 0.05 |
| Time In Bed Workdays (Sleep Duration Workdays) (hh:mm) | Evening-types | 07:40 | 01:15 | 07:58 | 01:12 | | | |
| | Intermediate-types | 08:10 | 00:55 | 08:01 | 01:20 | 0.30 | 0.74 | 0.02 |
| | Morning-types | 07:26 | 00:49 | 08:16 | 01:24 | | | |
| | Mean Survey Time | 07:46 | 00:59 | 08:05 | 01:19 | 2.58 | 0.12 | 0.07 |
| Time in Bed Free days (Sleep Duration Free days) (hh:mm) | Evening-types | 08:18 | 00:55 | 08:15 | 03:08 | | | |
| | Intermediate-types | 08:26 | 00:54 | 07:47 | 02:01 | 0.45 | 0.64 | 0.02 |
| | Morning-types | 08:29 | 01:31 | 08:56 | 01:33 | | | |
| | Mean Survey Time | 08:25 | 01:07 | 08:20 | 02:14 | 0.06 | 0.81 | 0.002 |
| **MidPoint of Sleep (hh:mm)** | Evening-types | 05:46 | 01:26 | 07:06 | 02:02 | | | |
| | Intermediate-types | 04:52 | 01:34 | 05:43 | 01:55 | 2.85 | 0.07 | 0.14 |
| | Morning-types | 03:27 | 01:26 | 06:11 | 03:25 | | | |
| | **Mean Survey Time** | **04:41** | **01:29** | **06:20** | **02:27** | **14.87** | **0.0001** | **0.29** |
| Social JetLag (hh:mm) | Evening-types | 01:28 | 01:02 | 02:02 | 03:34 | | | |
| | Intermediate-types | 01:51 | 00:57 | 01:29 | 02:34 | 1.42 | 0.26 | 0.07 |
| | Morning-types | 02:20 | 01:29 | 03:07 | 04:38 | | | |
| | Mean Survey Time | 01:53 | 01:10 | 02:13 | 03:35 | 0.25 | 0.62 | 0.007 |
| Passage of Present Time | Evening-types | 1.00 | 1.41 | 1.50 | 2.17 | | | |
| | Intermediate-types | 1.29 | 1.45 | 0.10 | 2.00 | 0.70 | 0.51 | 0.04 |
| | Morning-types | 0.63 | 1.60 | 0.38 | 2.33 | | | |
| | Mean Survey Time | 0.97 | 1.49 | 0.66 | 2.17 | 0.70 | 0.50 | 0.02 |
| Past Intervals | Evening-types | 1.25 | 0.57 | 0.88 | 0.94 | | | |
| | Intermediate-types | 0.95 | 0.82 | 0.68 | 0.90 | 0.86 | 0.43 | 0.05 |
| | Morning-types | 0.88 | 0.74 | 0.47 | 0.65 | | | |
| | Mean Survey Time | 1.03 | 0.71 | 0.68 | 0.83 | 4.32 | 0.045 | 0.11 |
| Life Periods | Evening-types | 0.73 | 0.83 | 0.19 | 0.93 | | | |
| | Intermediate-types | 0.59 | 0.74 | 0.53 | 0.54 | 1.41 | 0.26 | 0.07 |
| | Morning-types | 0.98 | 0.73 | 0.82 | 0.67 | | | |
| | Mean Survey Time | 0.77 | 0.77 | 0.51 | 0.71 | 2.83 | 0.10 | 0.07 |
| **Feeling of Time Pressure** | Evening-types | 2.78 | 0.64 | 1.96 | 0.98 | | | |
| | Intermediate-types | 2.61 | 0.78 | 2.32 | 0.89 | 0.54 | 0.59 | 0.03 |
| | Morning-types | 2.55 | 0.46 | 1.78 | 0.80 | | | |
| | **Mean Survey Time** | **2.65** | **0.63** | **2.02** | **0.89** | **22.42** | **0.0001** | **0.38** |
| **Feeling of Time Expansion/Boredom** | Evening-types | 1.46 | 0.85 | 1.82 | 0.77 | | | |
| | Intermediate-types | 1.28 | 0.70 | 1.85 | 0.83 | 0.77 | 0.47 | 0.04 |
| | Morning-types | 1.48 | 0.57 | 2.33 | 0.70 | | | |
| | **Mean Survey Time** | **1.41** | **0.71** | **2.00** | **0.77** | **20.32** | **0.0001** | **0.36** |
| Temporal Metaphor of Speed | Evening-types | 2.87 | 0.69 | 2.07 | 0.73 | | | |
| | Intermediate-types | 2.68 | 0.73 | 2.38 | 0.92 | 0.04 | 0.96 | 0.002 |
| | Morning-types | 2.50 | 0.98 | 2.50 | 0.69 | | | |
| | Mean Survey Time | 2.68 | 0.80 | 2.32 | 0.78 | 3.90 | 0.056 | 0.10 |
| Temporal Metaphor of Slowness | Evening-types | 1.67 | 0.96 | 1.43 | 0.93 | | | |
| | Intermediate-types | 1.16 | 0.40 | 1.75 | 0.77 | 0.16 | 0.85 | 0.01 |
| | Morning-types | 1.58 | 0.34 | 1.38 | 0.63 | | | |
| | Mean Survey Time | 1.47 | 0.57 | 1.52 | 0.78 | 0.07 | 0.79 | 0.002 |

In order to deeply address the significant results, the difference between the pre-lockdown period and the COVID-19 lockdown for each variable was calculated for every participant, as done by [25]. Thus, a positive difference indicated a reduction of value during the COVID-19 lockdown, whereas a negative difference indicated an increase of this value during home confinement. However, for the sake of clarity, for sleep and wake times during workdays, the difference was reversed (i.e., time during COVID-19 lockdown minus time during pre-lockdown period), and thus, a positive difference indicated a phase advance of time. Then, for the variables reporting a significant time effect in Table 1, the calculated (pre-lockdown minus COVID-19 lockdown) difference was tested against zero (i.e., whether the difference was significantly differed from zero). The differences for BTW (M = +02:00; SD = 03:19; $t(38)$ = 3.78, $p$ = 0.001), WTW (M = +02:10; SD = 03:31; $t(38)$ = 3.85, $p$ = 0.0001), MPoS (M = −01:22; SD = 02:29; $t(38)$ = −3.41, $p$ = 0.002), FTP (M = +0.52; SD = 0.78; $t(38)$ = 4.16, $p$ = 0.0001), and FTE/B (M = −0.57; SD = 0.76; $t(38)$ = −4.75, $p$ = 0.0001) were significantly different from zero. These results confirmed that during the COVID-19 lockdown, participants experienced a change of these variables, respectfully, to the pre-lockdown period.

To further examine the relationships between the individual changes over these two periods, a difference between values reported in pre-lockdown period and those obtained during the COVID-19 lockdown for each variable was calculated (i.e., positive differences indicated a reduction of values during social isolation, whereas negative differences indicated an increase of values in home confinement). For all bedtimes and wake-up times, the difference was calculated between COVID-19 lockdown times and pre-lockdown times (i.e., positive differences indicated a late shift for sleep–wake timing experience during the COVID-19 outbreak). Table 2 reports the correlation coefficients of all correlations between variables in all samples.

Beyond the fact that the rMEQ score did not correlate with any difference in values of every variable ($r$s ranged from −0.25 to +0.21 and $p$s > 0.12; correlations not reported in Table 2), Table 2 revealed 2 specific correlation patterns. Table 2 showed negative correlations between the dMAAS score and all (retrospective, prospective, and total) PRMQ scores, as well as the feeling of time expansion/boredom. Basically, more attention and awareness of the present moment during the pre-lockdown period, more subjective memory functioning was reported in home confinement, and a greater feeling of time expansion/boredom in the same period. Beyond expected correlations between variables—such as, for example, bedtime during workdays and MPoS or SJL, or dPT and dMoSpeed—no significant correlations between subjective memory functioning, sleep–wake habits, and/or subjective feeling of time passage were found.

Finally, the mediation model, hypothesizing an indirect effect of dispositional mindfulness on sleep timing during workdays, through the feeling of time expansion/boredom, was tested (Figure 1). The model tended towards the statistical significance (adjusted $R^2$ = +0.33, $F(5,33)$ = 3.31, $p$ = 0.015). Although the dMASS did not directly predict dBTW ($p$ = 0.32), dMAAS was negatively associated with dFE/B ($p$ = 0.007), which, in turn, was positively associated with dSTW ($p$ = 0.008), suggesting an indirect effect (−0.06, 95% CI = −0.16/−0.007). In other words, during the COVID-19 lockdown, less dispositional mindfulness induced a greater feeling of boredom, and this increased feeling of time expansion induced a late shift of bedtime during workdays.

**Table 2.** The correlation coefficients are reported above the major diagonal of the correlation matrix. For all variables, the difference between pre-lockdown and during COVID-19 lockdown was calculated, with the exception of sleep and wake times for whom the reversed difference was calculated. In bold are the significant correlations.

| | 1dMAAS | 2dRETRO | 3dPRO | 4dPRMQ | 5dBTW | 6dWTW | 7dBTF | 8dWTF | 9dTIBW | 10dTIBF | 11dMPoS | 12dSJL | 13dPT | 14dTI | 15dLP | 16dFP | 17dFE/B | 18dMo Speed | 19dMo Slow |
|---|---|---|---|---|---|---|---|---|---|---|---|---|---|---|---|---|---|---|---|
| 1 | 1 | −0.49 ** | −0.63 *** | −0.65 *** | −0.002 | +0.13 | +0.05 | +0.17 | −0.37 | −0.15 | −0.09 | −0.01 | −0.33 | +0.07 | −0.03 | −0.16 | −0.45 ** | −0.22 | −0.03 |
| 2 | | 1 | +0.54 *** | +0.82 *** | −0.05 | −0.17 | +0.14 | +0.24 | +0.36 | −0.15 | +0.07 | −0.01 | +0.15 | −0.18 | −0.10 | −0.04 | +0.17 | +0.06 | −0.01 |
| 3 | | | 1 | +0.92 *** | +0.03 | −0.10 | +0.10 | +0.06 | +0.39 | +0.01 | +0.02 | +0.06 | +0.30 | +0.02 | +0.07 | +0.03 | +0.27 | +0.19 | +0.03 |
| 4 | | | | 1 | +0.001 | −0.14 | +0.11 | +0.14 | +0.42 | −0.06 | +0.05 | +0.03 | +0.27 | −0.05 | −0.003 | +0.004 | +0.25 | +0.16 | +0.02 |
| 5 | | | | | 1 | +0.94 *** | +0.15 | +0.11 | +0.005 | +0.004 | −0.98 *** | −0.82 *** | +0.02 | −0.03 | +0.13 | +0.18 | +0.34 | −0.08 | +0.03 |
| 6 | | | | | | 1 | −0.02 | +0.07 | −0.34 | −0.09 | −0.96 *** | −0.77 *** | −0.04 | −0.006 | +0.19 | +0.11 | +0.19 | −0.16 | +0.13 |
| 7 | | | | | | | 1 | +0.49 ** | +0.48 ** | +0.29 | −0.21 | −0.15 | −0.09 | −0.19 | −0.25 | +0.01 | +0.34 | +0.02 | −0.20 |
| 8 | | | | | | | | 1 | +0.10 | −0.69 *** | −0.25 | +0.11 | −0.001 | +0.10 | −0.13 | +0.15 | −0.02 | +0.08 | *−0.31°* |
| 9 | | | | | | | | | 1 | +0.28 | +0.11 | +0.01 | +0.17 | −0.06 | −0.20 | +0.16 | +0.39 | +0.25 | −0.29 |
| 10 | | | | | | | | | | 1 | +0.10 | −0.24 | −0.07 | −0.27 | +0.07 | −0.15 | +0.30 | −0.07 | +0.17 |
| 11 | | | | | | | | | | | 1 | +0.78 *** | +0.02 | +0.02 | −0.12 | −0.16 | −0.28 | +0.11 | −0.03 |
| 12 | | | | | | | | | | | | 1 | −0.03 | −0.008 | −0.24 | −0.14 | −0.28 | +0.05 | −0.13 |
| 13 | | | | | | | | | | | | | 1 | +0.51 ** | +0.14 | +0.26 | −0.09 | +0.50 ** | −0.49 ** |
| 14 | | | | | | | | | | | | | | 1 | +0.13 | +0.36 | −0.20 | +0.59 *** | −0.28 |
| 15 | | | | | | | | | | | | | | | 1 | +0.23 | −0.16 | +0.04 | +0.05 |
| 16 | | | | | | | | | | | | | | | | 1 | −0.03 | +0.47 ** | −0.23 |
| 17 | | | | | | | | | | | | | | | | | 1 | −0.06 | +0.08 |
| 18 | | | | | | | | | | | | | | | | | | 1 | −0.30 |
| 19 | | | | | | | | | | | | | | | | | | | 1 |

To note that the letter d indicates difference. RETRO is retrospective memory; PRO is prospective memory; PRMQ is total PRMQ score; STW is sleep time during the workdays; WTW is wake-up time during workdays; STF is sleep time during free days; WTF is wake-up time during free days; TIBW is sleep duration during workdays; TIBF is sleep duration during free days; MoPS is mid-point of sleep; SJL is social jetlag; PT is present time; TI is temporal intervals; LP is life periods; FP is the feeling of time pressure; FE/B is the feeling of time expansion/boredom; Speed is the temporal metaphor of speed; and Slow is the temporal metaphor of slowness. In the Table, ** indicates $p < 0.005$, and *** indicates $p < 0.0005$.

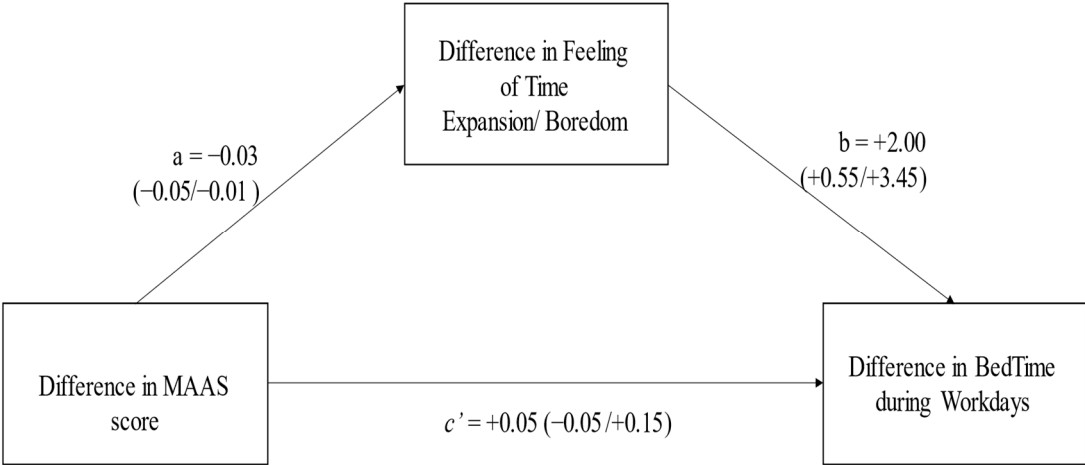

**Figure 1.** The mediation model tested the indirect effect (a and b parameters) between the difference of the MAAS score (positive values indicate a greater MAAS score in pre-lockdown) and the difference of bedtime during workdays (positive values indicate a late shift of sleep timing), through the mediation effect of the difference in the feeling of time expansion/boredom (positive values indicate less boredom during lockdown). The c' parameter indicates a direct effect. Within the brackets, the 95% CI of the effects.

## 3. Discussion

The general aim was to assess the relationships between the subjective passage of time, sleep timing/regulation, chronotype, dispositional mindfulness, and self-reported memories slips in a longitudinal study from the pre-lockdown period (December 2019) to the first Italian COVID-19 lockdown (from March to May, 2020). To address this general aim, in the present study the changes from pre-lockdown to home confinement were assessed for all variables. Then, the relationships between individual changes in these variables were addressed through correlational analysis. Finally, a possible mediation model explaining how changes in the dispositional mindfulness could predict the changes in the feeling of time extension/boredom, which, in turn, could be related to delayed bedtime during workdays during the COVID-19 pandemic, was tested.

Considering that this study could directly evaluate the impact of the home confinement in the passage of time, sleep–wake habits, morningness–eveningness preference, dispositional mindfulness and subjective memory functioning in the same participants, the present study showed that from December 2019 to March–May 2020 participants modified their sleep–wake (i.e., bedtimes and wake-up times) timing during the work/school days and in their feeling of time pressure and expansion. Although these findings partially confirmed the expectations, participants longitudinally reported an advanced shift for bed and wake-up times compared to those reported previously of social isolation [10,15,18,46–48]. This sleep timing was further corroborated by significant late occurrence of the MPoS, indicating that Italian individuals, probably during COVID-19 lockdown, followed their own biological rhythm [35], although no significant difference in rMEQ scores between pre-lockdown and lockdown periods was found, as well as the lack of any circadian typology effect on sleep–wake habits during the week (workdays and free days). Although the main circadian typology factor approached the significance, the reported delayed bedtime for evening-types, with respect to other remaining chronotypes during free days, seemed to support the previous assumptions and suggest that participants tended to follow their own biological clock in regulating the sleep–wake cycle during the social isolation. Even if in the present study no changes for sleep pressure [33] or sleep variability [34] during the home confinement were found, the late bedtimes and wake-up times, especially during the workdays, could be related to adverse health outcomes experienced during the COVID-19 lockdown [4–17], with particular influence for sleep health [10,15,18,46–48,66,67]. These

findings could have important public health implications, not only for further pandemic situations, for whom government countermeasures would be needed, but also for modern societies, in which irregular sleep is highly prevalent due to a chronic circadian disruption [79]. In addition, these findings highlight the importance of sleep timing and sleep regularity for sleep hygiene and sleep health, for sleep medicine, and psychology [35]. In regard to the subjective passage of time, two associated patterns were found: as the feeling of time pressure decreased, the feeling of time expansion/boredom increased during the COVID-19 outbreak [18–31]. These result patterns could indicate that the home confinement modified the daily schedule for all participants with the consequence of a subjective elapsing of time. In line with previous research, it was possible to interpret this distortion of the passage of time arguing for the emotional experience of everyday life during the lockdown [18–30]. In previous studies, it was reported an increased level of anxiety and depression (e.g., [66,67]), and thus it was possible that the perceived level of stress [80] or depression [19,20,23,25–27] was associated with a distortion of the passage of time, with an increase of boredom.

Contrarily from the expected correlational relationships, in the present study the negative associations between the reduction of the dispositional mindfulness and better retrospective and prospective memory, as well as the increase of the feeling of time expansion/boredom were only found. The former association described that a reduction of present-centered attention–awareness in everyday experience was related to a better memory functioning during social isolation. Although this relationship seemed to be counterintuitive, not only the present finding could be in line with mixed results on other cognitive tests [12,13,16,17], but it also suggested that the lack of attention to and awareness of the present moment gave space to travel in the past or in the future [57,58,60–64]. Although the present data did not clearly replicate the results found by Fiorenzato et al. [12], the mean PRMQ scores were greater when assessed during the COVID-19 lockdown compared to those measured during the pre-lockdown period in the same individuals. The reduction of dispositional mindfulness during lockdown could, in part, explain this paradoxical improvement of memory functioning due to, probably, a higher tendency to travel in the past and future times. The negative association between dispositional mindfulness and the feeling of time expansion/boredom suggested that a reduction of the MAAS score was related to a slowed down perception of time (and an increase of the feeling of boredom) during the COVID-19 lockdown, in line with previous assumptions [18–31]. In a similar way to meditators, the present data could indicate that the reduction of the trait or dispositional mindfulness determined a timelessness, with the consequence of a slowing down of the passage of time [62–64]. In addition, this relationship between mindfulness and subjective passage of time could confirm the relationship between the sense of time and the sense of body [60–64]. Indeed, it has been observed that the MAAS score was positively correlated with measures of openness, internal state, awareness, positive and pleasant affect, and well-being, and, on the opposite, it was negatively correlated with anxiety, stress, and rumination [81]. The changes of the mental health during the lockdown [4–17,66,67] could be related to low dispositional mindfulness, and, in turn, an increase of boredom with a slowing down of the passage of time or a feeling of time expansion. Future studies are needed to explore this hypothesized model both in healthy individuals and in meditation experts, using either a cross-sectional or longitudinal research designs.

The main novelty of this longitudinal study was related to the mediation model tested: the dispositional mindfulness exerted an effect on the feeling of time expansion/boredom, which, in turn, predicted the delayed bedtime during the workdays. Bearing in mind the strict relationship between the sleep timing (i.e., delayed bedtime) and sleep health, it is possible to advance the idea that the feeling of slowing down of time affected the sleep timing and sleep quality due to an increased boredom [19,24–27,31,66,67]. At the same time, a poor sleep quality "reinforced" the feeling of being bored and "determined" a delayed sleep–wake cycle in the subsequent day [28–32,68]. Considering that boredom has been associated with several problematic behaviors, such as bedtime procrastination

and phubbing [31], the present findings could indicate that mindfulness practice could decrease boredom [32], regulate the sleep–wake schedule [68] and sleep quality [32,65]. At the same time, the present mediation model seemed to be in line with previous findings [68] investigating the associations between mindfulness and sleep quality [66,67], as well as sleep timing and sleep quality [35], giving a complementary explanation of the impact of the COVID-19 lockdown on sleep quality [4–17,66,67]. However, the present study was not designed to deeply address the mechanisms of this interrelated relationship and/or in which way the changes in the dispositional mindfulness could modify the feeling of time expansion/boredom, allowing individuals to maintain a more regular sleep timing. Future studies should address these possibilities, probably focusing on the different mindfulness interventions in several clinical populations, such as, for example, Post Traumatic Stress Disorder (PTSD) patients who generally report altered sleep timing and regularity (e.g., [82]).

Altogether, the present findings indicated a possible way of the impact of home confinement—which was a government measure to limit the coronavirus spread—on the sleep disorders reported. Indeed, a reduction of the dispositional mindfulness could determine an increase of the feeling of time expansion/boredom (with associated negative feelings [19,24–30]), and, in turn, a delayed bedtime (or an increase of bedtime procrastination [31,32,68]) during the days in which the present sample was mainly engaged in remote work or university activities. Thus, mindfulness could act as an individual coping strategy for unpleasant situations and unpleasant feelings, usually associated with boredom. A reduction of feeling of time expansion/boredom could help individuals to maintain a regular sleep–wake schedule, probably because of a reduction of bedtime procrastination [31,32,49,68]. Indeed, bored individuals tend not to be successfully engaged in activity and/or perceive the current activity as meaningless, determining a search for a stimulating activity. This continuous search distracts them from paying attention to and being awarene of the present moment and from sleep, with a consequent increase of procrastination of bedtime in an attempt to find something to do [31,32,49,68]. In that sense, mindfulness practice could be a therapeutic intervention to increase its presence component with a reduction of negative emotions and feeling of time expansion/boredom, with a more regularity in sleep for bedtimes and wake-up times. Although, at the moment this article has been submitted, in Italy, as well as in other countries of the world, people have reported to cope and adapt to COVID-19 diffusion, and COVID-19 seems not to be an unsustainable pressure to the healthcare system (https://eurohealthobservatory.who.int/monitors/hsrm/ Accessed on 7 May 2023), it is not possible to exclude that a new variant of COVID-19 (or another type of virus; https://www.who.int/emergencies/disease-outbreak-news, Accessed on 7 May 2023) will determine a new widely contagion diffusion, requiring to re-adopt a severe lockdown. Indeed, in Europe, for example, there are different countries with a high rate of contagion (e.g., at the moment that this paper has been submitted, Bulgaria, Croatia, San Marino, etc.), and the mean number of cases in the last 7 days (at the moment of the paper has been written) was equal to 122,146, suggesting that COVID-19 remains an important topic for the health of the population (https://who.maps.arcgis.com/apps/dashboards/ead3c6475654481ca51c248d52ab9c61, Accessed on 7 May 2023). Thus, the present results could suggest that online training in mindfulness and home-based mindfulness practice could decrease individual boredom, increase the attention to the passage of present time, and reduce bedtime procrastination with a direct impact on sleep quality [65–67] and sleep timing [68]. In a more general sense, mindfulness could be proposed as a specific line of intervention, which country governments should adopt for coping with negative emotions and problematic behaviors, and for determining a sleep–wake regularity. Furthermore, these findings propose a possible answer to the question of how individuals could cope with the feeling of boredom and maintain a regular sleep timing in industrial societies, given that both of these aspects are related to health problems. Future studies should address this topic for the application of specific daily interventions, targeting populations at risk, such as adolescents [83].

However, this study has certain limitations that need to be considered. First, the sample size was small due to the adjustment of the original protocol study when the lockdown began. In March 2020, only a small group of people filled in the questionnaire and then agreed with the participation during home confinement. The choice to change the original protocol study did not allow to calculate an adequate sample size for this study by G*Power, and, probably, many null results, found especially when circadian typology was considered, could be related to a small number of participants generally, and of both extreme chronotypes, particularly. Moreover, the present study failed to show associations between changes in sleep timing/regulation and other variables, probably due to the small sample size. Contrarily from what was expected, the lack of significant correlations (Table 2) did not allow us to provide conclusive assumptions for the relationship between sleep and memories, between the subjective passage of time and sleep–wake timing, or between dispositional mindfulness and the sleep–wake cycle. However, as mentioned in the introduction, the merit of the study regarded the direct observation of the changes in several variables, from preceding the non-pandemic period to the pandemic situation. This aspect differed from many studies which were performed directly during the COVID-19 lockdown and required participants to compare their actual pandemic situation with a (remembered) period before the home confinement, with a high risk of memory bias. In addition, the choice to use a more conservative alpha level could protect the risk to a made-error in statistical decision, even if the sample size was small. Future studies should replicate the present findings with a large numerosity of participants, and possibly with an adequate number of evening- and morning-types in order to clarify the role of the circadian typology in explaining mixed results found in the literature, due to the association between evening-types and the distortion of the passage of time, more procrastination, and altered sleep timing and continuity [33,34,36–44]. At the same time, future studies should test the proposed mediation model changing the order of the variables in order to investigate alternative models with the aim to find strong evidence on a possible cause–effect relationship between these variables.

In line with this first limit, another aspect was related to the convenience sample recruited in this study because it limited representativeness and generalizability. At the same time, participants were not selected using inclusive and/or exclusive criteria in the first administration of the questionnaires, and then, with the change of the study from a cross-sectional to a longitudinal study due to the home confinement abruptly imposed by the governments, no further control criteria for the recruitment were adopted. However, the present study showed results in line with other studies with more representative samples with specific criteria for the selection of participants [12,18–32,66–68], and this convergence of the results could indicate that the sample of this research did not influence the data.

Another limit could be related to the use of self-report questionnaires. Although reliable and valid questionnaires were administered, the self-report measures are not exempt of problems, such as limited introspective abilities and social desirability. In addition, in the present study, any measures were taken to control for response bias or dishonest responses, considering that there was the possibility that during the second data collection, participants knew about the lockdown situation. However, it was possible to exclude the presence of potential biases in the response because participants were requested to indicate their opinion, feelings, habits, etc., limiting the possibility to give dishonest responses. In addition, the second data collection was performed from March (i.e., immediately at home confinement) to May (i.e., with still high contagion rate in Italy) 2020, and it was difficult to take into account that social desirability and/or response bias could affect the data, given that no one could know how to cope with this exceptional phenomenon. In addition, the second data collection was performed at least two months after the first administration of questionnaires. Future studies should adopt more objective and/or behavioral measures of the assessed variables (e.g., actigraphy for assessing sleep timing and regularity, or the creation of a waiting situation for manipulating the sense of time expansion/boredom), reducing the possible impact of subjective biases in the responses.

Related to this point, an additional limit could be related to the MAAS, which focused on the absence of attention to and awareness of present experience, for the assessment of the dispositional mindfulness. The Five Facet Mindfulness Questionnaire (FFMQ) or the Philadelphia Mindfulness Scale (PHLMS) [81] could be more adequate tools for assessing the trait mindfulness, because the FFMQ has five different facets, which could capture better the individual differences in dispositional mindfulness, and the PHLMS focuses on two different components of mindfulness—that is, awareness (a behavioral tendency of continuously monitoring current experience) and acceptance (a stance of experiencing events without judgements and reactions). These tools could cover more essential aspects of mindfulness with respect to the MAAS, which is a unidimensional scale. However, there is still debate about which mindfulness scale represents all the essential aspects of mindfulness, given that all mindfulness scales have merits and limits [81]. In addition, the MAAS, which is the most frequently used and evaluated tool, provides positive evidence for internal consistency, reliability, construct validity by hypothesis testing, and responsiveness [81]. Future studies should replicate the present study using a different mindfulness scale, with the possibility to indicate which mindfulness aspect is mainly related to time expansion/boredom and delayed bedtime.

## 4. Materials and Methods

### 4.1. Participants

As described in the study by Mirolli et al. [66], a convenience sample from the general population was recruited through emails, social media and personal contacts. During the pre-lockdown period, 43 volunteers participated in the survey, composed by paper-and-pencil questionnaires (see below). There were 23 males and 20 females, and the mean age was equal to 33.86 years (SD = 13.68 years). In this original sample, 4.70% obtained an eighth-grade diploma, 39.50% of participants obtained a high school diploma, 20.90% reported a bachelor's degree, and 32.60% reported a master's degree. The remaining 2.30% obtained a PhD title. All participants read the written consent form and agreed to take part in the study. The study was conducted according to the guidelines of the Declaration of Helsinki [84], and approved by the Ethics Committee of the Department of Psychology at the University of Campania Luigi Vanvitelli (protocol code Fabbri_5/2020). The study aimed in its first formulation to investigate the relationship between dispositional mindfulness chronotype, sleep–wake habits, memory functioning and time experience. However, when home confinement was introduced in Italy in response to the COVID-19 pandemic, the aim of the original study was adequate to this unexpected situation in order to assess the impact of the lockdown on all of the above variables. The reason for this shift from an initial cross-sectional study to a longitudinal study was based on the possibility to directly compare the data from preceding non-pandemic periods to those collected during the home confinement. The occupational status covered unemployment, university students, and workers in both private and public fields. Thus, all original participants were contacted again and asked to fill the same previous questionnaires considering the lockdown situation in an online survey using the Google Moduli platform. The choice to change the environmental survey (from a vis-à-vis collection to an online survey) was grounded on the severe limitations imposed by the Italian government with a severe social isolation. About 91% of participants (4 participants, corresponding to 9.30% of dropout, explicitly decided to not participate in this second online survey) of the original sample agreed to fill the questionnaires during the first Italian COVID-19 lockdown and the data were collected within the temporal window from 17 April to 10 May 2020. This final sample was composed of 21 men and 18 females, and the mean age was 35.03 years (SD = 14.02 years). There was not a significant difference between males (M = 36.76 years; SD = 15.41 years) and females (M = 33.00 years; SD = 12.32 years) for age ($t$(37) = 0.80, $p$ = 0.43). In this sample, 5.10% of participants obtained an eighth-grade diploma, 41.00% declared a high school diploma, 23.10% obtained a bachelor's degree, and 28.20% reported a master's degree. The remaining 2.60% of participants obtained a PhD title. There was

no gender difference for the educational level ($U$ = 142.50, $p$ = 0.17). All participants lived in the south of Italy, and the occupational status covered unemployment, students, and workers in both private and public fields. When the demographic characteristics of the original sample were compared with those of the final sample, no significant differences were found for age ($t(80)$ = −0.38, $p$ = 0.70), gender distribution ($\chi^2(1)$ = 0.001, $p$ = 0.97) and educational level ($U$ = 807.50, $p$ = 0.76), suggesting that the drop-out of participants did not influence the sample characteristics.

*4.2. Materials*

Dispositional mindfulness was assessed by the Mindful Awareness Attentional Scale (MAAS) [52]—for the Italian version of the MAAS, see [85]—which is one of the most widely used questionnaires to assess dispositional mindfulness [86]. The MAAS measures dispositional mindfulness using a unidimensional scale [52]. The MAAS is indicated to measure the attention component of mindfulness, focusing on the maintenance of awareness of the present moment experience [86]. The MAAS uses 15 items, measuring dispositional mindfulness (e.g., "*I rush through activities without being really attentive to them*"). Specifically, participants are requested to report how often they believed they had an experience referenced by each item on a scale from 1 (i.e., "*almost always*") to 6 (i.e., "*almost never*"). The total score of the MAAS involves calculating mean score across the 15 items, with higher averaged total score indicating greater mindfulness. In the present study, the Cronbach's alpha of the MAAS was equal to 0.85 and 0.90 for pre- and during lockdown, respectively.

The sleep–wake habits were assessed through four ad hoc questions [87] relating to what time participants usually go to bed (bedtime or BT) or wake-up (or WT) during workdays (W) or university days (usually from Monday to Friday), and free (F) days (usually corresponding to the weekend). These ad hoc questions allowed us to assess the typical sleep–wake habits of the participants with the possibility to calculate several sleep indices. Beyond the information of the habitual bedtimes and wake-up times, the Times In Bed (TIB or self-report sleep duration) during the workdays/university days and during free days were calculated as the time elapsed from BT to WT. Furthermore, the social jetlag (SJL) was calculated as the absolute difference between the mid-sleep point of sleep (MPoS) on free days and the mid-point of sleep during workdays [34]. The MpoS was defined as the middle time point between bedtime and wake-up timing. All these variables were expressed as hours:minutes.

The reduced version of the Morningness–Eveningness Questionnaire (rMEQ) [36]— for the Italian version of the rMEQ, see [88]—was administered to assess circadian typology. The rMEQ has demonstrated better psychometric characteristics than those reported by the original 19-item MEQ scale, requires short time for compilation with respect to MEQ, and is a reliable tool in chronobiological and chronopsychological research [36]. The 5-item scale derives from the original 19-item version of the MEQ. Three items ask about participants' preferred time for going to bed, getting up, and the hour of the day when peak personal efficiency is at its maximum. Moreover, participants assess their degree of tiredness within the first 30 min after awakening along a 4-point scale (1 = "very tired"; 4 = "very awake"), and to indicate which chronotype they think they belong to. The rMEQ score is obtained by summing the scores of each question. According to [88], a score of 4–10 corresponds to evening-types, 11–18 for intermediate-types, and 19–25 for morning-types. The Cronbach's $\alpha$ of rMEQ for the present study was equal to 0.56 and 0.57 for the first and second data collection periods, respectively. Despite low internal consistency, this tool has been widely used to assess circadian typology [89].

For cognitive performance, the Prospective and Retrospective Memory Questionnaire (PRMQ) [78] was used (at the following link an Italian version is available: https://www.ed.ac.uk/ppls/psychology/research/facilities/philosophy-and-psychology-library/psychological-tests/prmq). The PRMQ was selected because it is a reliable tool for assessing memory functioning in daily life [78]. In addition, it was previously used by Fiorenzato et al. [12],

and thus it was possible to assess the presence of the reported memory paradox during the COVID-19 pandemic. The PRMQ is generally used to assess memory slips that everyone can make in daily life. The questionnaire is a 16-item scale and participants have to indicate how frequently they experienced some retrospective (i.e., referred to the past; e.g., "*Do you fail to recognize a place you have visited before?*") and prospective (i.e., referred to the future; e.g., "*Do you decide to do something in a few minutes' time and then forget to do it?*") memory mistakes, ranging on a 5-point scale from 1 ("*never*") to 5 ("*very often*"). The total score can range from 16 to 80 and was obtained by summing the scores of each question. Considering that 8 items are used to assess retrospective memory errors and 8 items are used to assess prospective memory mistakes, both subscales can range from 8 to 16 as the result of the sum of score at each relative question. In line with Crawford et al. [78], the raw scores of the Prospective (Pro) and Retrospective (Retro) scales were transformed into T scores using the computer program available at www.psyc.abdn.ac.uk/homedir/jcrawford/prmq.htm. In this way, for total PRMQ score, as well as Pro and Retro subscales scores, higher scores indicated better memory functioning. The internal consistency of the PRMQ was equal to 0.85 and 0.90 for pre- and during COVID-19 lockdown period, respectively.

Finally, we adopted the Time Awareness and Subjective Time questionnaires (see Table 1 at page 925 in [90]) proposed by Wittmann and Lehnhoff [30,90]; for an Italian version of the questionnaires, see [22]. As stated by Wittmann and Lehnhoff [90], the time awareness can be defined as the subjective impression of time as moving quickly or slowly. The subjective passage of time indicates how quickly or slowly time seems to pass relative to a normal situation for an individual, and it can cover large time spans and does not require a comparison between subjective estimations and objective clock time [90]. The first part for the Time Awareness (TA) consisted of questions referred to the personal experience of time, and participants had to indicate how slowly or fast the time passes or has passed using a 5-point scale from −2 ("*very slowly*") to +2 ("*very fast*"). Specifically, the first two questions were related to the perception of present time, in the form of trait-like and ("*How fast does time usually pass for you?*") state-like ("*How fast do you expect the next hour to pass?*") perception of passage of present time (PT). In the present study, the sum of subjective assessment was calculated, and thus, positive values reflected a perception of a fast passage of present time, while negative values reflected a slow passage of present time. Then, four questions covering the retrospective judgments of the passage of past time intervals were proposed. Specifically, these questions asked how fast last week, last month, last year, and the past 10 years had passed, using the previous 5-point scale. In a similar way, participants had to indicate how fast their childhood (defined as the period before 12 years), adolescence (between 13 and 19 years), young adulthood (between 20 and 29 years) and adulthood (between 30 and 39 years) had gone by. For the purpose of the present study, the mean scores of subjective judgments of time intervals (TI) and those for life periods (LP) were separately calculated. As before, positive values indicated a fast passage of time whereas negative values indicated a slow passage of time. The second part of the questionnaire was used to assess the Subjective Time Experience, given that participants were requested to indicate their feeling of time pressure (TP, 5 items, e.g., "*I often think that time is running out*") or their feeling of time expansion/boredom (TE/B, 5 items, e.g., "*My time is not filled*"). To judge their feelings, for these 10 statements, participants used a 5-point scale from 0 ("*strong rejection*") to 4 ("*strong approval*"). For each feeling, a mean score was computed. In the last part of this questionnaire, participants judged how their time experience could be explained by three temporal metaphors of speed (MoSpeed, e.g., "*Time is a speeding train*") or by three temporal metaphors of slowness (MoSlow, e.g., "*Time is a quiet, motionless sea*"). As before, a mean score for each metaphor category was calculated.

### 4.3. Procedure

As stated in the paragraph 4.1, the original design of the present research was a cross-sectional study, and, during the pre-lockdown period, the questionnaires were administered individually, using a paper-and-pencil version of questionnaires after the informed consent

and an agreement for the participation were signed. When the design of the research was modified into a longitudinal study, the same participants from the first data collection were contacted again, using the same contact strategy adopted in December 2019. All participants who decided to continue the participation received the link with the questionnaires. Indeed, the questionnaires were administered online using Google Moduli, and it was stressed to think about their daily experience during the first Italian lockdown. During the online survey, informed consent and agreement to participate to the study were obtained by specific clicks provided in the survey. Then, participants provided their demographic characteristics and filled in the MAAS, the rMEQ, the ad hoc questions about the sleep–wake habits, and finally the Time Awareness and the Subjective Time. The order of the questionnaires was the same as adopted in the first administration.

### 4.4. Data Analysis

Statistical analyses were performed using SPSS software (IBM). First of all, I assessed the presence of possible changes in rMEQ scores between the two selected survey periods. In case of a null result, I decided to categorize participants as evening-, intermediate- and morning-types on the basis of their score in rMEQ obtained in the COVID-19 outbreak. Then, using a chi-squared test ($\chi^2$), a different gender distribution among chronotypes was assessed. In addition, Pearson and Spearman correlation analyses were performed to assess association, respectively, between age and educational level with an rMEQ score.

After assessing for the normality assumption of all scores, a set of mixed measures of ANOVAs, with Circadian Typology (3 levels), as a between-subjects factor, and Time (pre-lockdown vs. during lockdown) as a within-subjects factor, was run on each variable score. To further examine the relationships between the individual changes over time for all variables, I calculated the differences in scores between pre-lockdown and during COVID-19 shutdown and analyzed their correlations with Pearson correlation analysis. When significant Time effect was found in previous ANOVAs, a t-test against zero was performed in order to assess whether the changes in the score were significantly different from zero.

Finally, a mediation analysis assessed whether the changes in the feeling of time expansion/boredom mediated the relationship between the changes in the dispositional mindfulness and in the bedtime habit during the workdays, controlling for gender, age and educational level. A mediation analysis could use bootstrap analysis to evaluate the significance of indirect effects by using macro Process from SPSS developed by Hayes (Model 4) [91]. The bootstrapping procedure with 5000 bias-corrected bootstraps with 95% confidence interval (95% CI) was estimated [92].

The alpha level was equal to 0.01 in order to be more conservative in the statistical choice (see [44] for a similar procedure), given that multiple analyses and comparisons were performed.

## 5. Conclusions

This longitudinal study assessed the changes from pre-pandemic to COVID-19 lockdown for dispositional mindfulness, sleep timing, subjective passage of time and self-reported memory functioning. The present study showed a significantly delayed bedtime, wake-up time, and delayed MPoS, a decrease of time pressure and an increase in the feeling of time expansion/boredom. In addition, the changes in the dispositional mindfulness correlated negatively with the changes in scores of PRMQ, as well as the changes in the feeling of time expansion/boredom, suggesting that less attention to and awareness of the present moment induced better subjective memory functioning and a slowing down of the passage of time. More importantly, the present study reported that during the home confinement, the changes in the dispositional mindfulness predicted the increase of the feeling of time expansion/boredom which, in turn, predicted the delayed bedtime during workdays. Thus, this study showed how the presence component of the mindfulness was related to the subjective passage of time during social isolation, and this relationship could

delay the bedtime, which has been associated with several health outcomes [35]. Extending a previous study, reporting the relationship between mindfulness, boredom and bedtime procrastination [68], the present study highlighted how the changes in the sense of time in social isolation impacted sleep health and different psychological difficulties.

**Funding:** This research received no external funding.

**Institutional Review Board Statement:** The study was conducted in accordance with the Declaration of Helsinki and approved by the Institutional Review Board (or Ethics Committee) of the Department of Psychology (protocol code Fabbri_5/2020 approved on 4 February 2020).

**Informed Consent Statement:** Informed consent was obtained from all subjects involved in the study.

**Data Availability Statement:** The raw data supporting the conclusion of this article will be made available by the corresponding author upon request.

**Acknowledgments:** I would like to thank Clotilde Citino and Arianna Pupa for their help with data collection. In addition, I would like to thank all participants for their participation in this longitudinal study.

**Conflicts of Interest:** The author declares no conflict of interest.

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
