# Peer review of "Mindfulness, Subjective Cognitive Functioning, Sleep Timing and Time Expansion during COVID-19 Lockdown: A Longitudinal Study in Italy"

_2624-5175, doi:10.3390/clockssleep5020024_

Round 1

Reviewer 1 Report

The manuscript presents a timely and relevant longitudinal study investigating the changes in dispositional mindfulness, time experience, sleep timing, and subjective memory functioning during the COVID-19 lockdown in Italy. The novelty of the study lies in its comprehensive examination of multiple interconnected variables, providing valuable insights into the psychological and cognitive effects of prolonged periods of social isolation and restricted activities. Methodologically, the study appears to be well-designed, with appropriate measures employed to assess the key variables, although further clarification on participant recruitment and potential confounding variables would strengthen the paper.

The potential contribution of this study to the field of mindfulness research is adequate, as it sheds light on the potential benefits of mindfulness practice for mitigating the adverse effects of social isolation and disrupted routines. With some minor revisions and clarification of specific methodological details, the study promises to make a useful contribution to the field of mindfulness research. Please see more detailed comments below:

-        The introduction is too long. Please provide a much more concise overview of the relevant background.

-        The introduction could benefit from some improvements in terms of organization and clarity. The authors should provide clearer definitions of key concepts such as dispositional mindfulness and time experience, as well as better explain the distinction between retrospective and prospective memory components. Furthermore, the authors should streamline the discussion of previous research to avoid redundancies and make the text more concise.

-        Shouldn’t the second section be the Method?

-        It would be useful to know if any exclusion criteria were applied, and whether the sample is representative of the target population.

-        When describing the measures, the authors should mention if these questionnaires have been validated for the Italian population, especially if they were translated from English. Additionally, details about the procedure for obtaining the questionnaires' total and subscale scores could be helpful.

-        The MAAS has a lot of limitations as it is a measure of absentmindedness. It does not contain a range of facets, such as the FFMQ. This is a severe limitation that needs to be discussed.

-        The authors describe the pre- and during-lockdown data collection procedures using paper-and-pencil questionnaires and Google Moduli, respectively. It is important to note the potential for bias, as the participants knew about the lockdown situation during the second data collection period. The authors should also clarify if any measures were taken to control for response bias or dishonest responses.

-        The data analysis plan is comprehensive, utilizing SPSS for statistical analyses, and detailing the various tests and analyses to be performed. However, the authors should consider discussing assumptions for the mixed measures ANOVAs and the potential for violations of these assumptions. They should also address how they will handle missing data, if any, and if they plan to use any adjustments for multiple comparisons.

-        The section on limitations and recommendations for future research appears to be cut off, and should be expanded upon. The author should provide a comprehensive list of limitations in their study and provide suggestions for how future research could address these limitations or expand upon the current findings. Additionally, while the author does an excellent job discussing the results in relation to other literature, they should consider providing more examples of less directly related studies to further contextualize their findings.

The quality of English is fine.

Author Response

Reviewer#1 Clocks & Sleep:

The manuscript presents a timely and relevant longitudinal study investigating the changes in dispositional mindfulness, time experience, sleep timing, and subjective memory functioning during the COVID-19 lockdown in Italy. The novelty of the study lies in its comprehensive examination of multiple interconnected variables, providing valuable insights into the psychological and cognitive effects of prolonged periods of social isolation and restricted activities. Methodologically, the study appears to be well-designed, with appropriate measures employed to assess the key variables, although further clarification on participant recruitment and potential confounding variables would strengthen the paper”.

I would like to thank the reviewer#1 for her/his positive comments about the manuscript, regarding the novelty of the study and the methodology. I tried to clarify in this revised version of the manuscript the recruitment of participant and potential confounding.

The introduction is too long. Please provide a much more concise overview of the relevant background”.

I tried to shorten the introduction. In the original version of the manuscript, the introduction was long because I would like to be clear in identify all the relationships between variables. In this revised version, I tried to provide a more concise overview of the relevant background.

The introduction could benefit from some improvements in terms of organization and clarity. The authors should provide clearer definitions of key concepts such as dispositional mindfulness and time experience, as well as better explain the distinction between retrospective and prospective memory components. Furthermore, the authors should streamline the discussion of previous research to avoid redundancies and make the text more concise”.

I would like to thank the reviewer#1 for this comment. In the revised version of the paper, I tried to provide a clear definition of dispositional mindfulness as well as a better explanation of which is the main distinction between retrospective and prospective memory. However, I decided to delete from the text the expression time experience because it could be misunderstood and was not the focus of the paper. In addition, in the Method section I think that the clear description of time measures could help reader to understand the meaning of the concepts. In addition, I tried to shorten the introduction as requested without losing important information for the reader.

Shouldn’t the second section be the Method?

I usually agree with the reviewer#1 for this aspect. However, I used the Journal template file word for writing the manuscript and then uploading it. The format of this template provides the Method section at the end of the manuscript.

It would be useful to know if any exclusion criteria were applied, and whether the sample is representative of the target population”.

As regards this point, I did not apply any exclusion criteria with the exception related to who did not want to participate to the survey during the COVID-19 lockdown. As stated in the text in the participant section, this was a convenience sample who, for definition, is ad-hoc sample recruited by social media and personal contacts. In the participant section, I added a comparison between participants who started to fill in questionnaire in pre-lockdown period and those who remained and continued to participate to the survey during the COVID-19 lockdown. The only “exclusion criteria” were dropout to continue the participation.

When describing the measures, the authors should mention if these questionnaires have been validated for the Italian population, especially if they were translated from English. Additionally, details about the procedure for obtaining the questionnaires' total and subscale scores could be helpful”.

The reviewer#1 is right, and I clarified for each scale the Italian validated version of the tool. The only exception was for the PRMQ because, to best of my knowledge no psychometric validation of the PRMQ in Italian context has been provided. However, as stated in the paper, the authors who validated the PRMQ provide several already translated version of the PRMQ for several context, including Italian one. In addition, there is no doubt of a good translation of the Italian PRMQ because one of co-author (Sergio Della Sala) of the first validation of the PRMQ is Italian.

The MAAS has a lot of limitations as it is a measure of absentmindedness. It does not contain a range of facets, such as the FFMQ. This is a severe limitation that needs to be discussed”.

I do not completely agree with the reviewer#1 for the aspect that the MAAS has a lot of limitations given that, to best of my knowledge, is one of the most used questionnaires to measure dispositional or trait mindfulness. At the same time, I think that FFMQ has a lot of limitations, such as for example the fact that observe facet is poorly related to other psychological constructs and sometimes it is necessary to unify the nonjudging and nonreacting facets to capture specific component of mindfulness. Thus, I do not think that the choice to use MAAS instead of FFMQ is a severe limitation of the study. However, I acknowledged this aspect in the limitation section of the paper as requested.

The authors describe the pre- and during-lockdown data collection procedures using paper-and-pencil questionnaires and Google Moduli, respectively. It is important to note the potential for bias, as the participants knew about the lockdown situation during the second data collection period. The authors should also clarify if any measures were taken to control for response bias or dishonest responses”.

Although I thank the reviewer#1 for this point, I do not think that participants knew about the lockdown situation during the second data collection period. The COVID-19 disease abruptly interfered with the life of everyone and the scientific data, as well as the daily news, grew but not so linearly and constantly (with the addition of fake news), leaving us in uncertain situation. Although Italian government was quite fast in proposing countermeasures, the knowledge about the lockdown situation (and how to cope it) was not widespread, limiting any bias as suggested by the reviewer#1. In addition, participants performed this second data collection after a long-time window (about 1 or 2 months from their first participation) and, honestly, I doubt that they could remember something of the previous survey. However, I acknowledged the presence of potential bias in the discussion. To best of my knowledge, none of the used measure provides control items given that participants were requested to indicate their beliefs, daily experience, and habits. No control measures were adopted to control for response bias or dishonest responses, and I only added ad-hoc questions in the second data collection about the quarantine in line with previous published papers (e.g., which was the stress level of participants or whether they worked at home or not). However, I think that the great merit of this paper is related to the fact that it is a “true” longitudinal study given that I could assess the impact of lockdown procedure on different variables which were measured in a “safe” (without COVID-19 pandemic) period before the home confinement. Many papers in the literature compared the lockdown period with a “remembered” or “retrieved from memory” period when the COVID-19 did not exist. In these papers a potential bias related to memory bias is present, and I think this is not the case for the present study.

The data analysis plan is comprehensive, utilizing SPSS for statistical analyses, and detailing the various tests and analyses to be performed. However, the authors should consider discussing assumptions for the mixed measures ANOVAs and the potential for violations of these assumptions. They should also address how they will handle missing data, if any, and if they plan to use any adjustments for multiple comparisons”.

I thank the reviewer#1 for her/his positive comment about the result section. I added a statement about the normality assumption (from p= .064 for feeling of time expansion/boredom to p = .80 for MAAS score) of all variables for all analyses (ANOVAs and Pearson’s correlations) and I decided to modify the result section because I decided to fix the p value equal to .01 which is a more conservative alpha level for all results. In this way, the significant results should not be caused by type I error and at the same time a sufficient statistical power should be guaranteed (see the reviewer#2’s suggestion about the G*Power). Thus, many significant results of the original version of the paper disappeared and remained only results with a p value lower than .01. This procedure has been already adopted in previous papers (see reference 44 of the present study with further reference for this procedure).

The section on limitations and recommendations for future research appears to be cut off, and should be expanded upon. The author should provide a comprehensive list of limitations in their study and provide suggestions for how future research could address these limitations or expand upon the current findings. Additionally, while the author does an excellent job discussing the results in relation to other literature, they should consider providing more examples of less directly related studies to further contextualize their findings”.

I would like to thank the reviewer#1 for this point and for positive comment about the discussion. I expanded this section in the discussion and I hope that this part is now fine for the reviewer#1.

Comments on the Quality of English Language: The quality of English is fine”.

I would like to thank the reviewer#1 for this comment.

Reviewer 2 Report

The review results of this manuscript are as follows.

1.      Please check the description in line 14, that is, “39 Italian adults (21 horses; 35.03 ± 14.02 years)”. Describe the total number of study participants and the number of sex.

2.      What is the research question of this study? Please describe it in detail.

3.      Is the location of the materials and methods appropriate in this manuscript? Please check.

4.      Calculate the appropriate sample size for this study by G*Power.

5.      Selection criteria and exclusion criteria for participants in this study should be presented.

6.      Describe this research design.

7.      Describe the reason why the study participants were recruited only men.

8.      The findings should describe the demographic characteristics of the participants.

9.      There are only men in this study, and it's questionable how the results of the study suggested gender differences. Please check. Similarly, age and education level data were not presented, but did you calculate the difference in variables?

10.   Describe the survey environment and the surveyor.

11.   Describe the dropout rate and its cause in detail.

Author Response

Reviewer#2 Clocks & Sleep:

The review results of this manuscript are as follows.

“1.Please check the description in line 14, that is, “39 Italian adults (21 horses; 35.03 ± 14.02 years)”. Describe the total number of study participants and the number of sex”.

I would like to thank the reviewer#2 for her/his comment, and I checked the lines, and in the original abstract: “A longitudinal study was conducted on 39 Italian adults (21 males; 35.03 ± 14.02 years) assessing mindfulness, ad-hoc questions of sleep habits during workdays and free days, chronotypes, subjective time experience, and memory functioning, before (December 2019-March, 2020) and during (April 2020-May 2020) the first Italian COVID-19 lockdown”. So, no horses word appeared (probably horses derived from habits in the phrase below), the total number of study participants is reported (39) and the number of sex (21 males and 18 females; see the reference point below).

“2.What is the research question of this study? Please describe it in detail”.

I thank the reviewe#2 for this point and during the introduction and at the end of introduction I described the research question of the study (see also the relative comments of both other reviewers).

“3.Is the location of the materials and methods appropriate in this manuscript? Please check”.

I usually agree with the reviewer#2 for this aspect. However, I used the Journal template file word for writing the manuscript and then uploading it. The format of this template provides the Method section at the end of the manuscript.

“4.Calculate the appropriate sample size for this study by G*Power”.

I usually agree with the reviewer#2 for this aspect. However, in this case the sample size (which is equal to 78 participants, that is 39 participants assessed twice) derived from the unexpected event of the COVID-19 lockdown, which could not be foresaw in advance, adopting a more linear procedure to calculate the appropriate sample size for the study by G*Power. I think that the great merit of this paper is related to the fact that it is a “true” longitudinal study given that I could assess the impact of lockdown procedure on different variables which were measured in a “safe” (without COVID-19 pandemic) period before the home confinement. Many papers in the literature compared the lockdown period with a “remembered” or “retrieved from memory” period when the COVID-19 did not exist (and thus these studies have got the “time” to calculate an appropriate sample size). Thus, I I prefer to “sacrifice” the sample size (which, in my opinion is quite adequate for a longitudinal study) respect to the advantage of a “pure” picture of the changes produced by the home confinement. In any case, in order to reach a sufficient statistical power, I decided to put the alpha level equal to .01, that is a more conservative statistical level, limiting any problem related to an appropriate sample size. Thus, in the results section several significant results of the original paper were modified in no significant results in this revised version of the paper.

“5.Selection criteria and exclusion criteria for participants in this study should be presented”.

As stated in the participant paragraph the study was initially designed to investigate relationship between dispositional mindfulness chronotype, sleep-wake habits, memory functioning and time experience. Taken into account that all questionnaires require to indicate the beliefs, opinion, habits, etc of participants no selection criteria and exclusion criteria were defined at the beginning of the survey. With an abruptly change of the aim of the study (i.e., the changes of these variables during social isolation) the only “exclusion criteria” were the attritions/dropouts provided by initial participants who did not participate in the second part of the study.

“6.Describe this research design”.

I am sorry whether in the original version of the paper this aspect was not so clear. In the procedure section of the method, I described the research design (see also page 4 lines 180-188 for a description of the research design).

“7.Describe the reason why the study participants were recruited only men”.

I am sorry whether the reviewer#2 understood this erroneously aspect. As stated in the abstract (see relative point above), and in participant section of the method, the final sample (which participated to the longitudinal study) was made of 21 men and 18 women (for a total of 39 participants).

“8.The findings should describe the demographic characteristics of the participants”.

In this case I do not agree with the reviewer#2. In my opinion, the demographic characteristics of participants should be described in the participant section of the method and in the original version of the paper a description of these characteristics was present. However, I decided to expand this part reporting more information about the demographic characteristics of the participants.

“9.There are only men in this study, and it's questionable how the results of the study suggested gender differences. Please check. Similarly, age and education level data were not presented, but did you calculate the difference in variables?

As before, I am sorry for this misunderstanding with the reviewer#2. The study did not include only men. At the same time, age and educational level were already presented in the original version and also in this revised paper (lines 467-472). Although I can agree with the reviewer#2 that one single subject could have a birthday during the COVID-19 lockdown (and thus a change in age could be found), I do not think that this change was reliable (the same is true for educational level with a graduation during home confinement). Thus, no age or education level differences were calculated.

“10.Describe the survey environment and the surveyor”.

Although I think that it was sufficient to indicate that the second survey was performed via Google Moduli (to indicate the survey environment and the surveyor), I added a description related to what participants saw (and in which order) when they participated to online survey with Google Moduli.

“11.Describe the dropout rate and its cause in detail”.

As before, I am sorry for this misunderstanding. I think that the statement: About 91% of participants of the original sample agreed to fill the questionnaires and the data were collected within the temporal window from 17 April to 10 May 2020 was sufficient to describe the dropout rate. I specified the reason why I lost several subjects in this longitudinal study.

Reviewer 3 Report

I appreciate the opportunity to review the article Mindfulness, Subjective Cognitive Functioning, Sleep Timing

and Time Expansion During COVID-19 Lockdown: A

Longitudinal Study in Italy, about this work I have the following suggestions:

1.- It is recommended to adjust the introduction so that it is somewhat shorter.

2.- It would be convenient for the last paragraph to be organized into General Objective and Specific Objectives, and that the hypotheses be mentioned and listed in a scientific format, in order to later be able to analyze whether or not they are fulfilled.

3.- The methodology section is usually located before the results section, so when the results are read, it is understood how they were obtained. Unless the journal requires putting the methodology section at the end, it would be necessary to relocate it before the results section. In the methodology section, it would be convenient to explain why these instruments are chosen among the possible ones.

4.- The titles of tables and figures should be much shorter, one or two lines, at most.

5.- Although there is no problem that the text is written in the first person, it is not usual in scientific texts, the author can assess whether it is appropriate to change it.

Author Response

Reviewer#3 Clocks & Sleep:

I appreciate the opportunity to review the article Mindfulness, Subjective Cognitive Functioning, Sleep Timing and Time Expansion During COVID-19 Lockdown: A Longitudinal Study in Italy

I would like to thank the reviewer#3 for her/his positive comment.

“1.- It is recommended to adjust the introduction so that it is somewhat shorter”.

I thank the reviewer#3 for this comment. I tried to adjust the introduction, shorting it (see also the relative comment of the reviewer#1) and I hope that now all paragraphs of the introduction are focused on the presentation of the main data and for the rationale of the study.

“2.- It would be convenient for the last paragraph to be organized into General Objective and Specific Objectives, and that the hypotheses be mentioned and listed in a scientific format, in order to later be able to analyze whether or not they are fulfilled”.

I would like to thank the reviewer#3 for this suggestion. I re-wrote the last paragraph of the introduction with general and specific objectives of the study and the expected results. Although I did not list the hypotheses in a scientific format, I think that the expected results could help the reader to capture which results confirmed the expectation and which results did not.

“3.- The methodology section is usually located before the results section, so when the results are read, it is understood how they were obtained. Unless the journal requires putting the methodology section at the end, it would be necessary to relocate it before the results section. In the methodology section, it would be convenient to explain why these instruments are chosen among the possible ones”.

I usually agree with the reviewer#3 for this aspect. However, I used the Journal template file word for writing the manuscript and then uploading it. The format of this template provides the Method section at the end of the manuscript. I added statements for the reason why I chose these tools.

“4.- The titles of tables and figures should be much shorter, one or two lines, at most”.

The reviewer#3 probably is right. However, I think that the titles of tables and figures in the present study are informative about what the tables and figure displayed. I shortened them a little bit, and I hope that the reviewer#3 is satisfied.

“5.- Although there is no problem that the text is written in the first person, it is not usual in scientific texts, the author can assess whether it is appropriate to change it”.

The reviewer#3 is right, and I decided to delete the first person from the text and I made it in more impersonal way.

Round 2

Reviewer 2 Report

The results of the second review are as follows.

1.      According to the content presented in line 529 of the text, that is, "This final sample was combined by 21 men and 18 females," there are 18 female participants in this study. However, lines 13 to 14 of the abstract suggest "Alongitude study was conducted on 39 Italian adults (21 males; 35.03 ± 14.02 years)." As pointed out in the first review, confirmation of this is still needed.

2.      In this same view, it should be revealed whether the result values presented in the tables of this study are only for male participants or for male and female participants.

3.      If only male participants were included except for women, explanations for this should be described in detail because the time, quality of sleep, and sleep influencing factors of men and women are different.

4.      Describe why the difference values of the variables according to the demographic characteristics of the participants were not obtained during the two periods.

5.      There is a need for an operational definition of the dependent variables presented in this study.

6.      In addition, the dropout rate of participants in this study and the reason should be mentioned.

Author Response

Letter for the reviewers for the Manuscript ID Clockssleep-2363618, titled “Mindfulness, Subjective Cognitive Functioning, Sleep Timing and Time Expansion During COVID-19 Lockdown: A Longitudinal Study in Italy” submitted to section Society of Clocks & Sleep Journal.

Reviewer#2:

The results of the second review are as follows.

1.According to the content presented in line 529 of the text, that is, "This final sample was combined by 21 men and 18 females," there are 18 female participants in this study. However, lines 13 to 14 of the abstract suggest "Alongitude study was conducted on 39 Italian adults (21 males; 35.03 ± 14.02 years)." As pointed out in the first review, confirmation of this is still needed”.

Honestly, I think that it was quite intuitive that the total sample was made of 39 participants of whom 21 males and thus the remaining 18 females were derived from the subtraction 39-21. However, the reviewer#2 indicated this aspect in both rounds of the review and thus my intuition is wrong. Thus, I specified in the abstract the percentage of males to respect the word limit of abstract for the Journal.

2.In this same view, it should be revealed whether the result values presented in the tables of this study are only for male participants or for male and female participants”.

In the same view, I decided to specify this aspect.

3.If only male participants were included except for women, explanations for this should be described in detail because the time, quality of sleep, and sleep influencing factors of men and women are different”.

I hope that now this point was clarified. I did not include only men or only women, but I analysed all sample composed by men and women.

4.Describe why the difference values of the variables according to the demographic characteristics of the participants were not obtained during the two periods”.

I do not agree with the reviewer#2. In my opinion, in a longitudinal study, I do not think that age or educational level could be impacted by the lockdown. The difference values of analyzed variables were a way to assess the impact of the COVID-19 lockdown on these variables and they represented a change due to home confinement. As stated in the present response letter, it was probable that from December (first assessment) to March-April-May (second assessment), someone of participants had a birthday (at the same time, some students got a degree graduation) but I reported that the sample of the longitudinal study was similar to the initial sample for demographic characteristic. Thus, I do not think that I have to describe why I did not calculate the difference from first to second survey for demographic characteristics.

5.There is a need for an operational definition of the dependent variables presented in this study”.

For this point I do not agree with the reviewer#2. The material paragraph of the method section was improved from the original version of the manuscript due to the previous requests of all reviewers. I highlighted why I selected specific questionnaires for assessing specific variables, and this, in my opinion, is a sufficient way to define dependent variables operationally. In addition, the descriptions of all measures (and the way to obtain them) could be a sufficient way to report a operational definition of the dependent variables. Thus, in my opinion the material paragraph contained all information for an operational definition of the dependent variables presented in this study.

6.In addition, the dropout rate of participants in this study and the reason should be mentioned”.

As for the first round of the review process, the reviewer#2 required to specify this aspect and I wrote in the revised version of the paper the following statement: “About 91% of participants (4 participants, corresponding to 9.30% of dropout, explicitly decided to not participate in this second online survey) of the original sample agreed to fill the questionnaires during the first Italian COVID-19 lockdown and the data were collected within the temporal window from 17 April to 10 May 2020”. (lines 527-530).

Round 3

Reviewer 2 Report

This manuscript was properly revised according to the reviewer's comments. Thank you for your efforts.